# Verification of *Hypsibius exemplaris* Gąsiorek et al., 2018 (Eutardigrada; Hypsibiidae) application in anhydrobiosis research

**Izabela Poprawa**[1], **Tomasz Bartylak**[2,3], **Adam Kulpla**[4,5], **Weronika Erdmann**[2], **Milena Roszkowska**[3], **Łukasz Chajec**[1], **Łukasz Kaczmarek**[2], **Andonis Karachitos**[3], **Hanna Kmita**[3]*

**1** Institute of Biology, Biotechnology and Environmental Protection, Faculty of Natural Sciences, University of Silesia in Katowice, Bankowa, Katowice, Poland, **2** Department of Animal Taxonomy and Ecology, Faculty of Biology, Adam Mickiewicz University in Poznań, Uniwersytetu Poznańskiego, Poznań, Poland, **3** Department of Bioenergetics, Faculty of Biology, Adam Mickiewicz University in Poznań, Uniwersytetu Poznańskiego, Poznań, Poland, **4** Center for Advanced Technology, Adam Mickiewicz University, Uniwersytetu Poznańskiego, Poznań, Poland, **5** Faculty of Biology, Adam Mickiewicz University in Poznań, Uniwersytetu Poznańskiego, Poznań, Poland

* kmita@amu.edu.pl

## Abstract

Anhydrobiosis is considered to be an adaptation of important applicative implications because it enables resistance to the lack of water. The phenomenon is still not well understood at molecular level. Thus, a good model invertebrate species for the research is required. The best known anhydrobiotic invertebrates are tardigrades (Tardigrada), considered to be toughest animals in the world. *Hypsibius. exemplaris* is one of the best studied tardigrade species, with its name "*exemplaris*" referring to the widespread use of the species as a laboratory model for various types of research. However, available data suggest that anhydrobiotic capability of the species may be overestimated. Therefore, we determined anhydrobiosis survival by *Hys. exemplaris* specimens using three different anhydrobiosis protocols. We also checked ultrastructure of storage cells within formed dormant structures (tuns) that has not been studied yet for *Hys. exemplaris*. These cells are known to support energetic requirements of anhydrobiosis. The obtained results indicate that *Hys. exemplaris* appears not to be a good model species for anhydrobiosis research.

## Introduction

One of the most prevalent adaptations to water deficiency is anhydrobiosis, often called simply 'life without water', tolerance to desiccation or waiting for water to return [1–5]. More precisely, anhydrobiosis is described as the ability to dry to the point of equilibrium while exposed to moderately to very dry air (i.e., to 10% water content or even less) and then recover to normal functioning after rehydration without sustaining damages [6]. This denotes a series of coordinated events during dehydration and rehydration that are associated with preventing

**Data Availability Statement:** All relevant data are within the paper.

**Funding:** These studies were supported by the research grant of National Science Centre, Poland, NCN 2016/21/B/NZ4/00131. The funder had no role in study design, data collection and analysis, decision to publish, or preparation of the manuscript.

**Competing interests:** The authors have declared that no competing interests exist.

oxidative damages and maintaining the native structure at different levels of organism's organization [7, 8].

Anhydrobiosis is also described as an adaptation to unstable environmental conditions including drought or freezing, that allows the organism to survive when the environment becomes hostile to active life. Therefore, anhydrobiosis is considered to be a phenomenon of important applicative implications, enabling biostabilization and biopreservation as well as human disease treatment (e.g. [9–15]). Anhydrobiosis occurs in prokaryotes (e.g. [16]) and eukaryotes, with the latter including many microorganisms (e.g. [17]) as well as plants (e.g. [8]) and some small invertebrates (e.g. [18]). Among animals the best known example are tardigrades (e.g. [19]), indicated lately as an emerging source of knowledge of importance for medical sciences [13].

Tardigrade anhydrobiosis includes entry, dormant and exit stages, that correspond to the dehydration (i.e., tun formation), tun and rehydration stages, respectively [18, 20]. On the organismal level, the tun formation and return to the active stage have been quite well described and are understood fairly well [3, 21–26]. The key morphological changes during tun formation are longitudinal contraction of the body, invagination of the legs and intersegmental cuticle that are then reverted during rehydration. However, responsible cellular and molecular mechanisms are not yet fully described.

At the present, the genomes of only two tardigrade species are available i.e. *Hypsibius exemplaris* Gąsiorek, Stec, Morek & Michalczyk, 2018 [27] (in earlier works misidentified as *Hys. dujardini* (Doyère, 1840) [28] and *Ramazzottius varieornatus* Bertolani & Kinchin, 1993 [29–32], both representing the eutardigrade lineage [33]. The genomes enabled identification of proteins significant for tardigrade anhydrobiosis including some intrinsically disordered proteins regarded as unique for tardigrades (for review, see [26, 34, 35]). Moreover, both genomes allowed for comparative transcriptomics that corroborates experimental data indicating that different evolutionary tardigrade lineages may exhibit unique physiological and molecular adaptations to survive anhydrobiosis [36]. Accordingly, *Ram. varieornatus* is regarded as more tolerant to anhydrobiosis than *Hys. exemplaris* [13, 26, 37–39]. Nevertheless, *Hys. exemplaris* is one of the best studied tardigrade species, with its name "*exemplaris*" referring to the widespread use of the species as a laboratory model for various types of studies, ranging from developmental and evolutionary biology, through physiology and anatomy to astrobiology (e.g. [27, 40–44]).

It is frequently suggested that *Hys. exemplaris* requires a period of preconditioning to mobilize protectants needed to undergo a successful anhydrobiosis. However, the available protocols are based on different time windows and values of relative humidity (RH) for the preconditioning and dehydration. They also differ in the applied walking surface substratum as well as rehydration process and the reported levels of recovery following rehydration ranging between ca. 22 and 100% (e.g. [37, 39, 40, 45–47]). The second approach consists in slow dehydration under conditions of decreased RH but the recovery is not stated [48]. Therefore, we decided to verify the anhydrobiotic capabilities of *Hys. exemplaris*, which is crucial for the species applicability as a model in research of anhydrobiosis. For this purpose, we tested three different protocols, i.e. the protocol based on preconditioning, published by [49], our own protocol that we also use for other tardigrade species [50] and based on slow dehydration as well as a third one we termed "environmental drying" applied in two variants, using moist fine sand or a pond sediment as substrates. Fine sand and pond sediments are natural habitats for freshwater tardigrades including *Hys. exemplaris* [23, 27, 51], which motivated our decision to use of them as inorganic and organic drying substrates respectively. The obtained results indicate that in *Hys. exemplaris* anhydrobiosis, slow dehydration may be a better strategy than preconditioning. However, despite being a useful model in studies of other aspects of tardigrade

biology, *Hys. exemplaris* appears not to be a good model for anhydrobiosis research because of the process of storage cells' degeneration in tuns.

## Materials and methods

### *Hypsibius exemplaris* rearing

*Hypsibius exemplaris* Z151 strain (Fig 1) was purchased from Sciento (Manchester, United Kingdom) in 2015. To maintain the culture, specimens were kept in POL EKO KK 115 TOP+ climate chamber (photoperiod 12h light/12h dark, 20˚C, relative humidity (RH) of 50% on Petri-dishes (55 mm in diameter) with their bottoms scratched using sandpaper to allow movement of tardigrades. They were covered with a thin layer of the culture medium obtained by mixing double-distilled water and spring water (Żywiec Zdrój S.A., Poland) in ratio of 3 to 1. *Chlorella vulgaris* Beijerinck 1890 [52] (SAG211-11b strain) was served as a food once per week after the dish cleaning. Animals were transferred to a new culture dishes every few months (for details see [53]). The algae strain was kindly provided by Marcin Dziuba (Department of Hydrobiology, Faculty of Biology, Adam Mickiewicz University, Poznań, Poland) and was obtained from the culture collection of algae (Sammlung von Algenkulturen (SAG)) at the University of Göttingen, Germany.

### Anhydrobiosis protocols

For tun formation, fully active (displaying coordinated movements of the body and legs) adult *Hys. exemplaris* specimens of medium body length (approximately 200–250 μm) were extracted from the culture. After removal of debris, tun formation was performed using three different protocols, designated as A, B and C. In all protocols tardigrades were starved for one day before the protocol beginning. In protocol A, provided by Boothby [49], specimens were transferred onto 2% agar-coated lids of Petri dishes of 3.5 cm in diameter, in the minimal amount of the culture medium. The lids, termed "agar plates", were transferred for 16 h to a humidified chamber with RH 92%, obtained by application of 10% glycerol solution in a small plastic box with a lid (Fig 1). After the preconditioning, the agar plates were transferred to POL EKO KK 115 TOP+ chamber and kept in 40% RH for 24 h. Then, obtained tuns were kept in a desiccator for 7 days at 22% RH. All stages of tun formation were performed at controlled temperature of 20˚C. Protocol B consisted in application of slow dehydration of specimens by transferring them into 3.5 cm (in diameter) covered and vented Petri dishes with filter paper CHEMLAND 150 (06-00A102.150) placed on their bottom (Fig 1). Specimens were transferred in 400 μl of the culture medium and were left to dry slowly in the Q-Cell incubator (40–50% RH, 20˚C) for 72 h. The obtained tuns were kept in the incubator for 7 days. Protocol C, termed "environmental drying" was applied in two variants, i.e. C1 and C2. In both variants specimens were placed, together with 400 μl of the culture medium, into 3.5 cm (in diameter), covered Petri dishes containing ca. 5 ml of previously autoclaved (121˚C, 20 minutes, 100 kPa) substrate and were left to dry in Q-Cell incubator (40–50% RH, 20˚C) for 72 h. The dishes were kept in incubator for 7 days following drying of the substrate. In protocol C1 the substrate consisted of terrarium fine sand (Vitapol), with ca 3 ml of the culture medium added to moisturize it, while in protocol C2 sediment collected from a pond near the Faculty of Biology, Adam Mickiewicz University in Poznań, Poland (52˚ 28' 7.3956"N; 16˚ 56' 1.356"E), containing soil and decomposing plant matter was used as the substrate. The numbers of specimens and repeats used for estimation of survival rate as well as substratum for each of the applied protocols are summarized in Table 1. In the case of the A and B protocols, each plate contained five additional specimens for tuns' photography under stereomicroscope and tuns' ultrastructure analysis under transmission electron microscopy (see below). In the

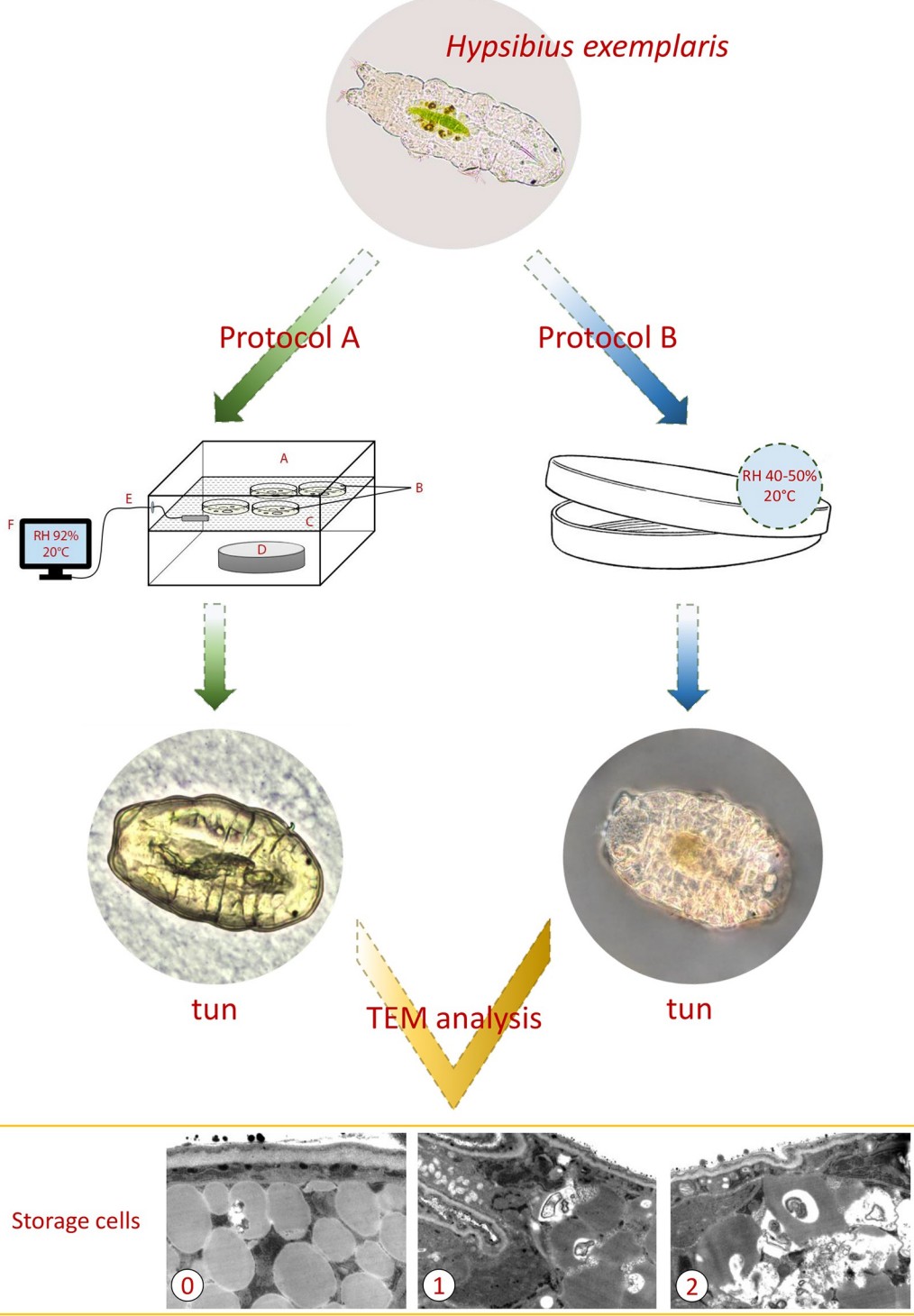

**Fig 1. Scheme of the experimental setup of A and B protocols used for ultrastructural analyzes.** Protocol A is represented by the sketch of preconditioning procedure and protocol B by a plate used for slow dehydration. **A**. small plastic box with lid; **B**. 2% agar-coated lids of Petri dishes ("agar plates"); **C**. scaffold for agar plates; **D**. a glass watch dish containing 10% glycerol solution; **E-F**. digital hygrometer; TEM, transmission electron microscopy; 0, 1 and 2, the distinguished three stages of degeneration of storage cells in typical tuns. High definition images obtained using TEM analysis, with relevant scale bars added, are presented in Fig 3.

**Table 1. Summary of the applied anhydrobiosis protocols.** In the case of A and B protocols, each plate contained five additional specimens for tuns' photography under stereomicroscope and tuns' ultrastructure analysis under transmission electron microscopy.

| Protocol | Quantitative details | Mode of dehydration | Substratum |
|---|---|---|---|
| A. | 10 repeats, each for 20 individuals | preconditioning | agar layer |
| | (20 specimens per plate) | | |
| B. | 5 repeats, each for 50 individuals | slow dehydration | filter paper |
| | (50 specimens per plate) | | |
| C1. | 5 repeats, each for 50 individuals | environmental drying | moist fine sand |
| | (50 specimens per plate) | | |
| C2. | 5 repeats, each for 50 individuals | environmental drying | sediment from the pond |
| | (50 specimens per plate) | | |

case of the C1 and C2 protocols, microscopic analysis of tuns was impossible due to the applied walking surface substratum which made observation and extraction of tuns impossible.

The animals' survival rate after 24 h following rehydration was observed in small glass cubes under the stereomicroscope (Olympus SZX7 and SZ51). In case of A and B protocols, the rehydration was performed by addition of 2 ml of the culture medium to each dish. Tuns were then transferred from their dishes to separate glass cubes and kept at 20 ˚C, and 40–50% RH. In the case of C1 and C2 protocols, contents of each dish were placed in larger Petri dish filled with culture medium to allow extraction of animals to the separate glass cube kept at 20 ˚C, and 40–50% RH. Successful survival was defined as the presence of coordinated movements of animal body and legs (crawling). Statistical significance of results was tested using unpaired t-test.

## Tun microscopic analysis

Tun formation by application of protocols A and B was observed under an Olympus SZ61 stereomicroscope connected to Olympus UC30 microscope digital camera. Randomly selected representative tuns were photographed on agar plates using the stereomicroscope. Tuns obtained by protocol B before transferring to agar plates were fixed in solution of 2.5% glutaraldehyde in 0.1 M sodium phosphate buffer (pH 7.4) at room temperature for 2 minutes. The fixation of the tuns was necessary to extract them from the filter paper fibres. Tardigrades penetrate the filter paper fibres during tun formation and the tuns cannot be removed without the filter paper being wet. Thus, the use of a fixative prevents rehydration of specimens and allows removal of the tuns from the dried filter paper.

Out of tuns obtained by protocols A and B, 10 tunes were randomly selected for each of the protocols to perform ultrastructure analysis under transmission electron microscope (TEM). Tuns were selected just before rehydration and then fixed in 2.5% glutaraldehyde prepared in 0.1 M sodium phosphate buffer (pH 7.4, 4˚C, 24 h), postfixed in 2% osmium tetroxide in 0.1 M phosphate buffer (pH 7.4, 4˚C, 1.5 h), dehydrated and embedded according to the protocol by [41]. The material was cut into ultrathin (50 nm) sections on a Leica Ultracut UCT25 ultramicrotome. These sections were mounted on formvar covered copper grids, stained with uranyl acetate and lead citrate, and analyzed with use of a Hitachi H500 transmission electron microscope at 75 kV.

## Results and discussion

### Applied anhydrobiosis protocols result in formation of correct tuns, but differ in survival rate

The applied anhydrobiosis protocols differed from each other at tun formation (dehydration procedure) while the rehydration procedure was similar (Table 1). As shown in Fig 2,

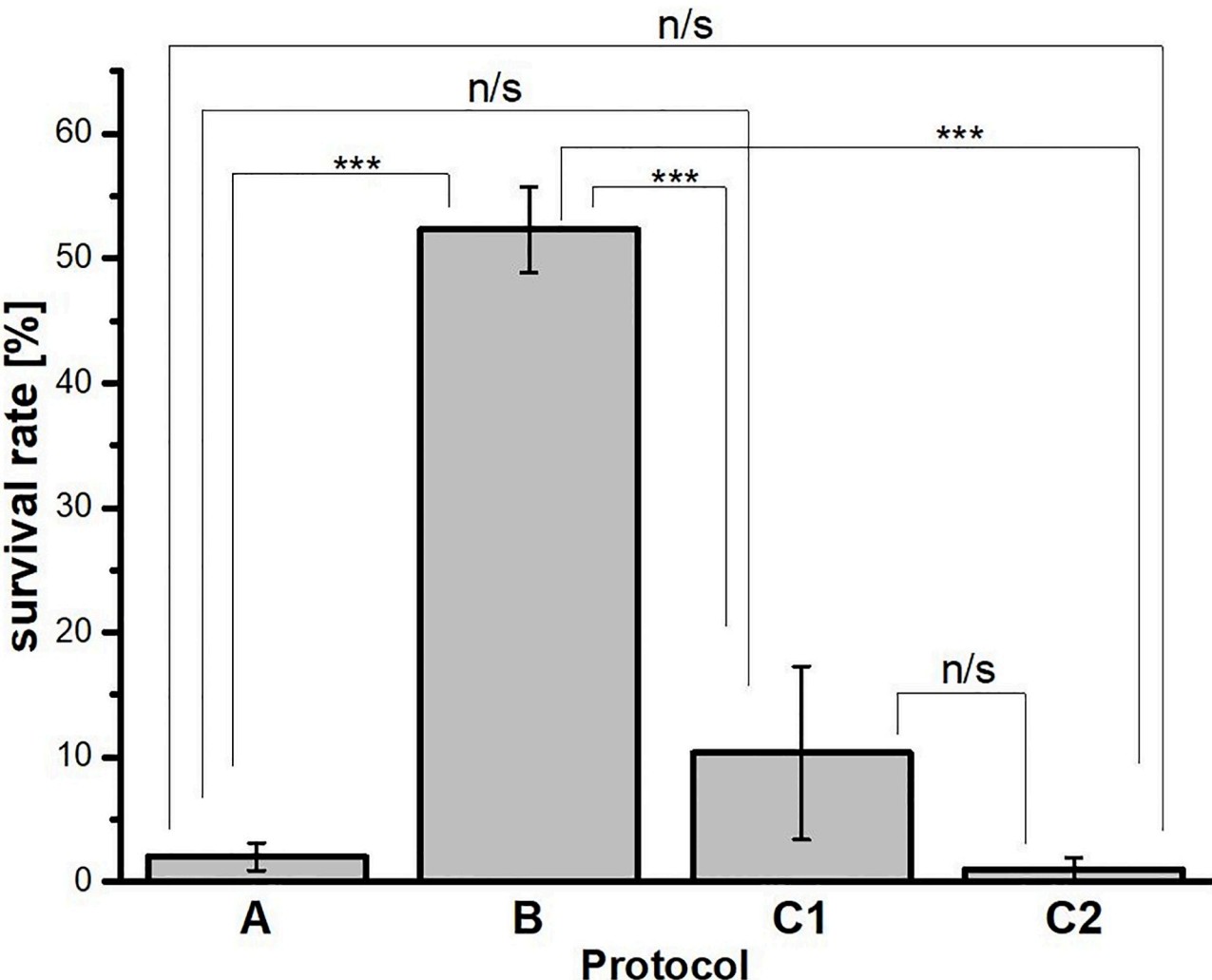

**Fig 2. Survival rate of *Hys. exemplaris* specimens after 7 days in tun stage.** The survival rate corresponds to percentage of specimens able to return to full activity after 24 h following rehydration. A, B, C1 and C2—symbols assigned to applied anhydrobiosis protocols. A, preconditioning on agar; B, slow dehydration on filter paper; C1, environmental drying in moist fine sand; C2, environmental drying in pond sediment. Data represent mean values ± SEM (see also Table 1). ***p < 0.001; n/s, not statistically significant.

reasonable survival rate, defined as coordinated movements of the body and legs (crawling) after 24 h following rehydration, was observed for protocol B (slow dehydration on filter paper). In the case of protocol C1 (environmental drying in sterile moist fine sand) survival was variable whereas in the case of protocol A (preconditioning on agar layer) and C2 (environmental drying in pond sediment) survival was very low. To explain these differences in survival rate we decided to check the appearance of formed tuns. It should be mentioned that for C1 and C2 protocols, microscopic analysis of tuns was impossible due to the applied walking surface substratum which made observation and extraction of tuns impossible. As shown in the research scheme presented in Fig 1, protocols A and B led to contraction of the body and withdrawal of legs into the body cavity accompanied by loss of water from the body, resulting in a distinctly shrunken body shape and the body size between ca.100-130 µm. The latter is in the range of the body compaction observed for *Hys. exemplaris* tuns [47]. Thus, the protocols allowed for formation of tuns with typical appearance. However, the typical appearance did

not guarantee successful return to full activity after rehydration following 7 days spent in a tun stage. Thus, typical appearance of tuns cannot be regarded as indicative of successful anhydrobiosis for *Hys. exemplaris* specimens. Therefore, we decided to analyse ultrastructure of ten randomly selected typical tuns obtained both by A and B protocols.

## Typical tun appearance does not rule out degeneration of storage cells at ultrastructural level

Ultrastructural analysis of tuns was performed by transmission electron microscopy (TEM). The analysis was based on storage cells, which are regarded as useful for ultrastructure analysis aimed to estimate possibility of successful return of tuns to active life, due to the cells being described as a factor affecting survival rates during anhydrobiosis [54]. Accordingly, the cells are described as the main form of energy storage, enabling a proper nutrient regime for different tissues as well as providing protection to tissues by producing protective metabolites [19, 55, 56]. Moreover, it has been shown that energetic support is also important during the tun stage [57]. Therefore, we assumed that tuns that were to survive should have had the correct ultrastructure of these cells, enabling their proper function also during/after rehydration. Out of tuns obtained by protocols A and B, 10 tuns were randomly selected for each of the protocols. The obtained TEM images of storage cells allowed to assign three stages of degeneration of these cells in typical tuns: 0—cells with no signs of degeneration; 1—cells with the first signs of degeneration; 2—cells with highly advanced degeneration (Fig 3 and Table 2). In stage 0, the storage cells had oval or ameboid shape and their electron-dense cytoplasm was filled with spheres of reserve material. Between the spheres, ribosomes, cisterns of rough endoplasmic reticulum, and shrunken mitochondria with electron dense matrix were visible (Fig 3A and 3B). As stated by Richaud et al. [47], mitochondria in tuns appear to have shorter cristae of the inner mitochondrial membrane and because oxidative phosphorylation occurs mostly in the deeply invaginated cristae it can be assumed that autophagy allows tardigrades to survive starvation. Accordingly, autophagosomes were observed sporadically in stage 0 (Fig 3A). In stage 1, the storage cells had the same shape and ultrastructure as described for stage 0, but in their cytoplasm, single vacuoles and autophagosomes appeared (Fig 3C). Accordingly, in the distinguished stage 2, the storage cells underwent severe vacuolization, and in their cytoplasm numerous autophagosomes were observed, and some of the autophagosomes were also disintegrated (Fig 3D and 3E). Moreover, the cell membrane of some cells was degraded (Fig 3E) and mitochondria had electron lucent matrix (Fig 3D). The latter is observed for damaged mitochondria with impaired functionality (e.g. [58]), which may result in cell death. All the stages were observed for storage cells in tuns obtained by protocol B, i.e. out of 10 analysed tuns, five displayed features of stage 0, two of stage 1, and three of stage 2, whereas for storage cells in tuns obtained by protocol A only stage 2 was observed (Table 2). Interestingly, these observations correlated with the survival rate determined for tuns obtained by A and B protocols (Fig 2). Thus, it could be concluded that in the case of protocols A and B, tardigrades forming tuns without visible degeneration of storage cells appear to be able to successfully return to active life. Thus, it can be assumed that the state of storage cells in tuns could be indicative of successful anhydrobiosis. Nevertheless, ultrastructure analysis of other cell types would strengthen this conclusion, i.e. digestive cells, epidermal cells and oocytes.

According to our knowledge, it is the first report indicating possibility of degeneration of storage cells in *Hys. expemplaris* tuns of typical appearance, resulting in their decreased survival. Available data on *Hys. exemplaris* tuns of comparable duration [47] concern only tuns of classical cellular structure. Moreover, the functional state of anhydrobiotic *Hys. exemplaris* storage cells has not been studied yet although the cells are known to accumulate polysaccharides and

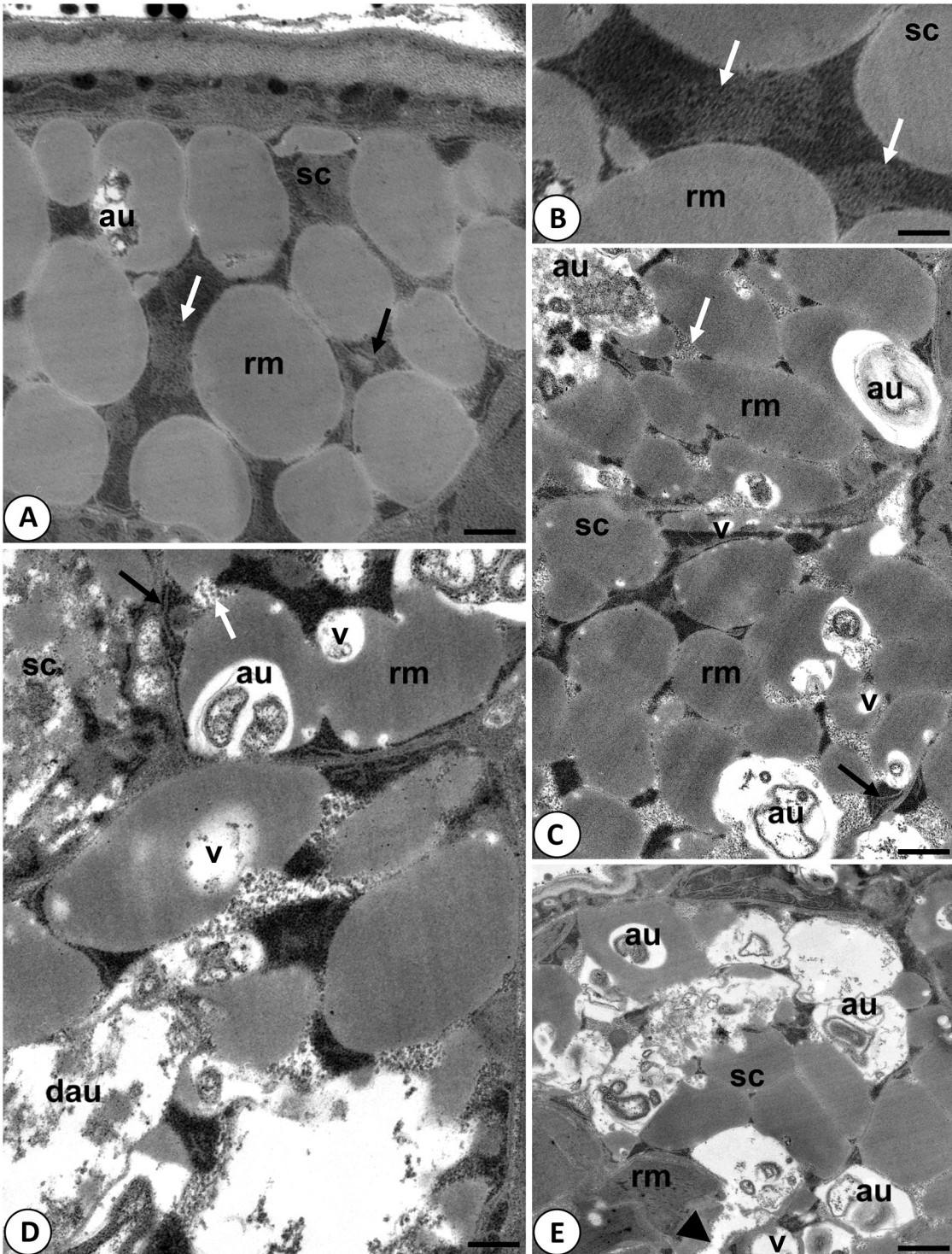

**Fig 3. Ultrastructure of storage cells in tuns of *Hys. exemplaris*. A-B**. Storage cells in stage 0—cells with no signs of degeneration: au–autophagosome rm- reserve material, sc—storage cell, black arrow—cisternae of rough endoplasmic reticulum, white arrow—mitochondrion; **A**. scale = 0.38 µm; **B**. scale = 0.23 µm; **C**. Storage cells in stage 1– cells with the first signs of degeneration: au–autophagosome rm- reserve material, sc—storage cell, v- vacuole, black arrow—cisternae of rough endoplasmic reticulum, white arrow—mitochondrion; scale = 0.41 µm; **D-E**. Storage cells in stage 2– cells with highly advanced degeneration: au–autophagosome, dau—disintegrated autophagosome, rm- reserve material, sc—storage cell, v–vacuole, black arrow—cisternae of rough endoplasmic reticulum, white arrow—mitochondrion, arrowhead–degraded cell membrane; **D**. scale = 0.29 µm; **E**. scale = 0.56 µm (see also Table 2).

**Table 2. Summary of ultrastructural analysis of storage cells in typical tuns.** For the analysis, 10 tuns obtained by protocol A and 10 tuns obtained by protocol B were randomly selected.

| Protocol | degeneration stage | Description | Approx. percentage [%] |
|---|---|---|---|
| A | 2 | highly advanced degeneration | 100 |
| B | 0 | no signs of degeneration | 50 |
| | 1 | first signs of degradation | 20 |
| | 2 | highly advanced degeneration | 30 |

lipids (e.g. [42]), and to be related to anhydrobiosis success because of their role of energy supplier (e.g. [19, 55]). We can assume that the observed damage to mitochondria may distinctly impair their role. However, it should be mentioned that in different tardigrade species the effect of anhydrobiosis on storage cells may be different as reflected by differences in changes of storage cells' size observed after dehydration [56]. Moreover, our data indicate that preconditioning is not a necessary element of *Hys. exemplaris* anhydrobiosis protocol as slow dehydration appears to provide even a better outcome. It should be remembered that application of different definitions of *Hys. exemplaris* recovery from the tun stage may hinder the comparison of the applied protocol effectiveness. For example there is a difference between "We defined recovered animals as those exhibiting spontaneous movements or at least responding to touch stimuli"[37] and the approach applied in this report, i.e. "coordinated movements of the body and legs (crawling)". Additionally, some of the available papers do not contain clear definition of the recovery (e.g. [46]), estimation of survival rate [48] or the indication of time window for survival estimation following rehydration [39] as well as duration of anhydrobiosis [37, 39, 45, 48].

Summing up, *Hys. exemplaris* is able to form tuns of typical appearance, but the process of storage cells degeneration decreases the tun survival distinctly. Thus, the species does not appear to be a good model in anhydrobiosis research.

## Acknowledgments

Studies have been conducted in the framework of activities of BARg (Biodiversity and Astrobiology Research group). The technical contribution of Kamil Janelt is highly appreciated.

## Author Contributions

**Conceptualization:** Łukasz Kaczmarek, Hanna Kmita.

**Data curation:** Łukasz Kaczmarek.

**Formal analysis:** Izabela Poprawa, Adam Kulpla, Milena Roszkowska, Łukasz Kaczmarek, Hanna Kmita.

**Funding acquisition:** Hanna Kmita.

**Investigation:** Izabela Poprawa, Tomasz Bartylak, Adam Kulpla, Weronika Erdmann, Milena Roszkowska, Łukasz Kaczmarek, Andonis Karachitos.

**Methodology:** Izabela Poprawa, Milena Roszkowska, Łukasz Kaczmarek.

**Project administration:** Łukasz Kaczmarek.

**Resources:** Łukasz Kaczmarek.

**Supervision:** Łukasz Kaczmarek, Hanna Kmita.

**Validation:** Izabela Poprawa, Milena Roszkowska, Łukasz Kaczmarek, Hanna Kmita.

**Visualization:** Izabela Poprawa, Milena Roszkowska, Łukasz Chajec, Andonis Karachitos, Hanna Kmita.

**Writing – original draft:** Izabela Poprawa, Tomasz Bartylak, Milena Roszkowska, Łukasz Kaczmarek, Andonis Karachitos, Hanna Kmita.

**Writing – review & editing:** Izabela Poprawa, Tomasz Bartylak, Milena Roszkowska, Łukasz Kaczmarek, Andonis Karachitos, Hanna Kmita.

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
