## [Decision Letter · Decision Letter 0]

10 Jan 2022

PONE-D-21-37709Verification of Hypsibius exemplaris Gąsiorek et al., 2018 (Eutardigrada; Hypsibiidae) application in anhydrobiosis researchPLOS ONE

Dear Dr. Kaczmarek,

Thank you for submitting your manuscript to PLOS ONE. After careful consideration, we feel that it has merit but does not fully meet PLOS ONE’s publication criteria as it currently stands. Therefore, we invite you to submit a revised version of the manuscript that addresses the points raised during the review process.

Dear Authors, two external reviewers have now assessed your manuscript "*Verification of Hypsibius exemplaris Gąsiorek et al., 2018 (Eutardigrada; Hypsibiidae) application in anhydrobiosis research*”, providing the comments that are reported below. As you can see, they both found identified a number of issues that would require careful revision before this paper is recommendable for acceptance.

Based on the reviewers' assessment, I'm thus here inviting you to take all of these comments into careful consideration and to modify your manuscript according to the provided constructive suggestions. I will then be happy to receive and further examine your revised version together with a point-by-point reply to each comment by myself and each reviewer, where you will need to explain any changes done to a particular piece of text, or include supported and convincing counterarguments to any points you may disagree with I'm confident you will find the present comments and suggestions relevant and useful to improve your work and I'm thus looking forward to hearing back form you by the due time.

We look forward to receiving your revised manuscript.

Kind regards,

Marcos Rubal García, PhD

Academic Editor

PLOS ONE

Journal Requirements:

[These studies were supported by the research grant of National Science Centre, Poland, NCN 2016/21/B/NZ4/00131. The technical contribution of Kamil Janelt is highly appreciated.

Studies have been conducted in the framework of activities of BARg (Biodiversity and Astrobiology Research group).]

 [These studies were supported by the research grant of National Science Centre, Poland, NCN 2016/21/B/NZ4/00131.]

Reviewers' comments:

Reviewer's Responses to Questions

**Comments to the Author**

1. Is the manuscript technically sound, and do the data support the conclusions?

Reviewer #1: Partly

Reviewer #2: No

2. Has the statistical analysis been performed appropriately and rigorously? 

Reviewer #1: I Don't Know

Reviewer #2: No

3. Have the authors made all data underlying the findings in their manuscript fully available?

Reviewer #1: Yes

Reviewer #2: Yes

4. Is the manuscript presented in an intelligible fashion and written in standard English?

Reviewer #1: Yes

Reviewer #2: Yes

5. Review Comments to the Author

Reviewer #1: This manuscript describes several alternative ways to dehydrate tardigrades of the species Hypsibius exemplaris. The alternative means used are interesting but the manuscript lacks precision in the motivation of these experimental choices. The quality of the images presented does not fully support the conclusions presented in the manuscript and needs to be improved.

Reviewer #2: General comments:

In this manuscript, authors used the tardigrade Hypsibius exemplaris and three dehydration protocols to study the shape of the tuns, the ultrastructure of the storage cells and the survival during rehydration.

After seven days, the three anhydrobiosis protocols used in this study showed variable efficiency in the formation of viable tun. The transmission electron microscopy analysis of tuns specimens showed storage cells degeneration status. Those status can be correlated with variable efficiency in the formation of viable tun. Propawa et al. concluded that only tuns without visible degeneration of storage cells are able to support a successful rehydration process.

This manuscript needs major revisions. Indeed, the poor quality of the pictures doesn’t allow to conclude anything.

Specific comments:

Text:

1. Line 84: There is no reference or accurate description of the experimental protocol for “environmental drying”. Why did authors choose these two substrates?

2. Line 96: Propawa et al. used "double-distilled water and Spring water" instead of the classical Chalkley medium (http://cshprotocols.cshlp.org/content/2018/11/pdb.prot102319.full). Please justify this choice.

3. Line 108: In this work, authors mentioned Boothby’s protocol (http://cshprotocols.cshlp.org/content/2018/11/pdb.prot102327.full.pdf). In this protocol, tardigrades are starved a day before dehydration. In the present work, it’s not described? Please specify if a starvation was conducted.

4. Line 125: Propawa et al. described a terrarium sand substrate.

Line 128: Authors described a substrate consisting of soil and decomposed plants.

I have the same remark as point 1. Why did they choose these two substrates?

5. Line 148 + Line 151: Authors mentioned twice the same fixation protocol with 2.5% glutaraldehyde. Only one is enough.

6. Line 150 to Line 153: Protocols A and B are totally different.

Tuns obtained by protocol A were photographed on agar plates using an Olympus SZ61 stereomicroscope.

Tuns obtained using protocol B were fixed in 2.5% glutaraldehyde in 0.1 M sodium phosphate buffer, mounted in a drop of water on a slide, covered by coverslip, and photographed with the use of Olympus BX60 stereomicroscope and OLYMPUS DP50 camera.

It is impossible to compare both specimens.

7. Line 154: Authors indicated that 10 specimens were analyzed. Is it 10 specimens in total or 10 specimens per experimental condition? It is not clear. Indeed, line 131, 5 specimens are mentioned.

8. Lines 175 to 177: Propawa et al. describe Figure 1. As indicated in another comments (comment on figure 1 and point 6), the quality of these pictures is not sufficient to compare the two protocols and protocols are not comparable. A figure 1 modification is required.

9. Line 181: Same question as the point 7. Please clarify the number of tardigrades.

10. Line 186: TEM ultrastructural analysis of specimen integrity is based only on storage cells according to a previous study in another species (Czernekova, 2018). Is the analysis of a single cell type sufficient?

11. Line 192: Authors describe mitochondria in Figure 3A. But the quality of the picture needs to be improved. This figure doesn’t allow to observe the state of the mitochondria or mitochondria cristae. We can’t conclude anything on mitochondria with this figure.

12. Line 194: Authors describe for Figure 3B: "single vacuole and autophagosomes appeared". But, according to me, we can also observe a vacuole in stage O.

13. Line 198: Like above, a better image quality is required to allow conclusions.

14. Line 212: " ca” should be replaced by "can".

Figures:

Figure 1: The quality of the two tuns pictures is insufficient. Images are totally different and cannot be compared to each other. As described in the M&M section, two different stereomicroscopes are used. The exposure for the two pictures is also different. The picture for the protocol B doesn’t allow to correctly observe the tun. The resolution is not sufficient.

For the storage cells TEM pictures, we don’t have scale bar.

Figure 3: The quality of pictures needs to be improved. Status of mitochondria or cristae for exemple are not clearly distinguishable.

Figures captions:

Figure 1: A description of tuns in the middle of the figure is required. And it’s precisely pictures that cannot be compared.

Figure 2: Statistical t-test is necessary. The test is mentioned in the M&M section but we can’t find the result in the rest of the paper.

Figure 3: Scale bars are very different between pictures (0.38um, 0.53um, 0.42um,…). For an easy and good reading of images, it is necessary to homogenize the scale bars.

6. PLOS authors have the option to publish the peer review history of their article (what does this mean?). If published, this will include your full peer review and any attached files.

Reviewer #1: No

Reviewer #2: No

---

## [Author Response · Author response to Decision Letter 0]

21 Feb 2022

Dear Editor,

All our comments and corrections to the revised manuscript are included into attached files.

Kind Regards,

Łukasz Kaczmarek

---

## [Decision Letter · Decision Letter 1]

28 Feb 2022

Verification of Hypsibius exemplaris Gąsiorek et al., 2018 (Eutardigrada; Hypsibiidae) application in anhydrobiosis research

PONE-D-21-37709R1

Dear Dr. Kaczmarek,

We’re pleased to inform you that your manuscript has been judged scientifically suitable for publication and will be formally accepted for publication once it meets all outstanding technical requirements.

Kind regards,

Marcos Rubal García, PhD

Academic Editor

PLOS ONE

Additional Editor Comments (optional):

Reviewers' comments:

Reviewer's Responses to Questions

**Comments to the Author**

1. If the authors have adequately addressed your comments raised in a previous round of review and you feel that this manuscript is now acceptable for publication, you may indicate that here to bypass the “Comments to the Author” section, enter your conflict of interest statement in the “Confidential to Editor” section, and submit your "Accept" recommendation.

Reviewer #1: All comments have been addressed

Reviewer #2: All comments have been addressed

2. Is the manuscript technically sound, and do the data support the conclusions?

Reviewer #1: Yes

Reviewer #2: Yes

3. Has the statistical analysis been performed appropriately and rigorously? 

Reviewer #1: Yes

Reviewer #2: Yes

4. Have the authors made all data underlying the findings in their manuscript fully available?

Reviewer #1: Yes

Reviewer #2: Yes

5. Is the manuscript presented in an intelligible fashion and written in standard English?

Reviewer #1: Yes

Reviewer #2: Yes

6. Review Comments to the Author

Reviewer #1: (No Response)

Reviewer #2: Authors have addressed all my comments and remarks. They increased explanations concerning their experimental choices. They also improved the quality of images as suggested. In its present form, the paper can be accepted for publication.

7. PLOS authors have the option to publish the peer review history of their article (what does this mean?). If published, this will include your full peer review and any attached files.

Reviewer #1: No

Reviewer #2: No

---

## [Editor Report · Acceptance letter]

10 Mar 2022

PONE-D-21-37709R1 

Verification of *Hypsibius exemplaris* Gąsiorek et al., 2018 (Eutardigrada; Hypsibiidae) application in anhydrobiosis research 

Dear Dr. Kaczmarek:

I'm pleased to inform you that your manuscript has been deemed suitable for publication in PLOS ONE. Congratulations! Your manuscript is now with our production department. 

Kind regards, 

on behalf of

Dr. Marcos Rubal García 

Academic Editor

PLOS ONE